evolution/computational biology/theoretical biology

evolutionary dynamics, fitness landscapes and complex systems

**Author for correspondence:**
Paulo R. A. Campos
e-mail: paulo.acampos@ufpe.br

# Analysis of statistical correlations between properties of adaptive walks in fitness landscapes

## Sandro M. Reia[1] and Paulo R. A. Campos[2]

[1]Instituto de Física de São Carlos, Universidade de São Paulo, Caixa Postal 369, 13560-970 São Carlos, São Paulo, Brazil
[2]Evolutionary Dynamics Lab, Physics Department, Federal University of Pernambuco, Recife, Brazil

PRAC, 0000-0002-7823-1998

The fitness landscape metaphor has been central in our way of thinking about adaptation. In this scenario, adaptive walks are idealized dynamics that mimic the uphill movement of an evolving population towards a fitness peak of the landscape. Recent works in experimental evolution have demonstrated that the constraints imposed by epistasis are responsible for reducing the number of accessible mutational pathways towards fitness peaks. Here, we exhaustively analyse the statistical properties of adaptive walks for two empirical fitness landscapes and theoretical NK landscapes. Some general conclusions can be drawn from our simulation study. Regardless of the dynamics, we observe that the shortest paths are more regularly used. Although the accessibility of a given fitness peak is reasonably correlated to the number of monotonic pathways towards it, the two quantities are not exactly proportional. A negative correlation between predictability and mean path divergence is established, and so the decrease of the number of effective mutational pathways ensures the convergence of the attraction basin of fitness peaks. On the other hand, other features are not conserved among fitness landscapes, such as the relationship between accessibility and predictability.

## 1. Introduction

In evolutionary biology, adaptive processes may be studied with the aid of fitness landscapes. This concept establishes a relation between the genotype of an individual and its reproductive success such that higher fitness represents best-suited specimens [1,2]. As a consequence, adaptive landscapes can be viewed as rugged surfaces with hills and valleys, and selective pressures drive the evolution of individuals towards the genotypes

placed in peaks. In this context, natural evolution is depicted by an uphill movement in the genotype configuration space of an evolving population [3].

This hill-climbing metaphor for adaptive evolution has inspired a huge body of theoretical works, including proposals of fitness landscape models [4–6]. The topography of the fitness landscape determines the distribution of selection coefficients and the amount and strength of epistasis [7]. Epistasis is essentially a measure of the deviation from additive assumption of fitness effects [8], which means that individual locus alterations may result in non-trivial fitness changes due to the correlations between the genetic loci of a genotype. The existence of local maxima is the hallmark of a non-additive landscape and hence a signature of epistasis [9]. In a seminal work, where a genetic reconstruction of the protein $\beta$-lactamase was accomplished, Weinreich demonstrated that the number of accessible paths towards fitter genotypes is remarkably smaller than the ensemble of possible trajectories owing to epistasis [10,11], and so evolution seems to be much more predictable and reproducible than previously expected [12].

Massive parallel evolution experiments and advances in the next-generation sequencing has allowed the assessment of a large amount of evolutionary information on empirical landscapes [13–15], though their analyses have been restricted to small parts of the landscapes and their inferred topography may not be truly representative [16]. Another limitation of those empirical analyses relies on the fact that the topography itself is shaped by the environment, which in turn is likely to change during evolution [17–19]. In this perspective, theoretical landscape models became a helpful tool to forecast evolutionary dynamics and produce replicates of fitness landscapes that preserve global patterns of epistasis [16]. Within this category, the NK model, introduced by Kauffman and Weinberger [5,20,21], seems to reasonably capture some of the features of empirical fitness landscapes [12,22]. Thus, the study of both empirical and theoretical fitness landscapes are complementary.

Evolutionary adaptation is driven by the accumulation of beneficial mutations, i.e. those that confer a fitness increase. The process is greatly influenced by the rate at which beneficial mutations arise, and also their selective effects [23–25]. In the simplistic view of evolutionary adaptation, namely strong-selection weak-mutation regime, the conditions $NU \ll 1$ and $Ns \gg 1$ hold, where $N$ stands for population size, $U$ is mutation rate and $s$ the selective advantage conferred by the beneficial mutations. Those conditions ensure that selection proceeds much faster than mutations occur. According to this picture, the population is monomorphic most of the time, and the dynamics can be approximated by an adaptive walk [5,24,26,27], in which the population is depicted as a single entity that moves through the fitness landscape towards fitness peaks. In short, adaptive walks arise as an idealized behaviour in the so-called strong-selection weak-mutation regime of more general dynamics such as the Wright–Fisher model [24,27,28]. The state of the system is described by the genotype $S$ shared by the population and moves are only allowed to the set of fitter mutant neighbours $\{G_i\}$, in which the Hamming distance $d(S, G_i) = 1$.

There are different versions of adaptive walks found in literature, and the choice of the version to be studied depends on the details of how the strong-selection weak-mutation is obtained. In the simplest version of the adaptive walk problem, dubbed random adaptive walks, the amplitudes of fitness differences are ignored, and so all fitter one-mutational step neighbours are equally likely to be chosen [26,29,30]. A second dynamics is referred to as natural adaptive walks, and now the selective effects play a role in such a way that the higher the mutant's fitness the higher the chance the population jumps to it [28]. Thus, for each $G_i$ in $\{G_i\}$, $s_i(G_i)$ is defined as the difference $s_i = f_i - f$, where $f$ is the fitness of the wild-type sequence $S$, and $f_i$ denotes the fitness of its one-mutational step neighbour $G_i$. Hence, the probability of choosing the sequence $G_i$ as the target sequence is given by

$$P_i(s_i) = \frac{s_i}{\sum_j s_j}. \tag{1.1}$$

The sum runs over all neighbours satisfying the condition $s_j > 0$. The natural adaptive walk here referred to as the probabilistic adaptive walk falls halfway between the random adaptation walk and the greedy adaptation, in which the population always moves to the fittest neighbour, and mimics adaptive processes under a large mutational supply [28,31,32]. The greedy adaptation walk is therefore deterministic. Both versions here addressed, the random and probabilistic adaptive walks, are very similar concerning the behaviour of the mean walk length, the accumulated number of beneficial mutations to a local optimum, which is expected to grow as $\ln L$ with sequence size $L$.

Within a theoretical perspective, the adaptive walk problem has been extensively investigated and several statistical properties assessed, such as the mean walk length, walk length distributions, distributions of fitness values at the ending points of the walks, and so on. Those theoretical works have mostly considered uncorrelated fitness landscapes [5,29,30]. Studies of correlated fitness

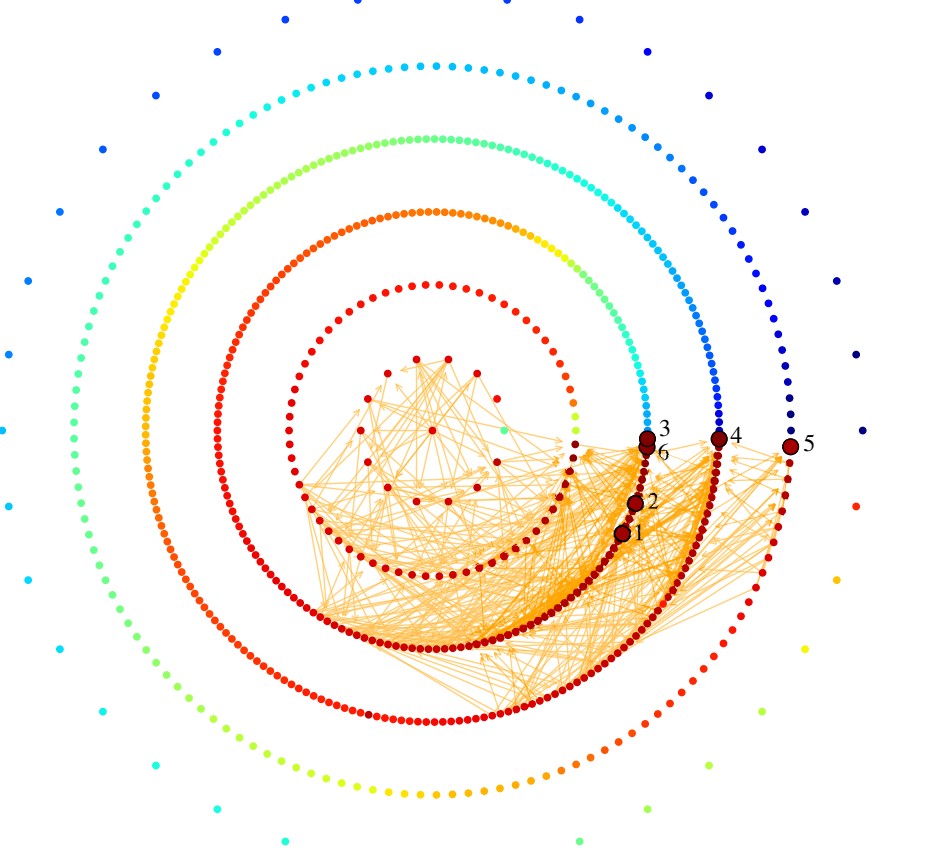

**Figure 1.** The graph comprising the 640 engineered mutants in the heat-shock protein Hsp90 in Saccharomyces cerevisiae. These mutants correspond to all possible combinations of 12 amino-acid changing mutations at six sites. Each node represents a given sequence, the wild-type (QFGWSANME) is symbolized by the central node, whereas the numbered nodes denote the six local maxima of the fitness landscape, namely, (1) QFGWTPAME, (2) QFGLTALME, (3) QFGFSALTE, (4) QFGLSPLAE, (5) QFGLTPAQE and (6) QFGISALQE. The innermost circumference amounts to all sequences that are one mutational step away from the wild-type sequence. The second smallest circumference encompasses those nodes that are two mutational steps away from the wild-type sequence, and so on. Accordingly, most of the local maxima are three mutational steps away from the wild-type sequence, and the global maximum is given by sequence (4). The nodes are coloured according to the fitness of the configuration, from dark blue (lowest fitness) to dark red (highest fitness), and the node-position in a given layer is given by its relative fitness: from $\theta = 0$ to the layer's node with the lowest fitness to $\theta \rightarrow 2\pi$ to the layer's node with the highest one. All possible mutational transitions between configurations are given by the directed links shown in orange. The data to generate the plot were obtained from Bank *et al.* (PNAS, 2016).

landscapes unveil that the topography of the landscape plays a role, usually resulting in longer walk lengths when compared to the uncorrelated case but this cannot be stated as a general rule [26,33,34]. The dependence of the adaptive walk length on the initial fitness has also been addressed. While some studies show that walk lengths seem to decrease with increased initial fitness [26,34,35], on the contrary, some experimental studies are compatible with a scenario of no dependence of the walk length on the initial fitness, with the assertion that mutations of large fitness difference are expected in the case of poorly adapted populations [36,37].

Here, we extensively survey the statistical properties of adaptive walks in two empirical fitness landscapes, Hsp90 and Gb1, and in the theoretical NK fitness landscapes. The Hsp90 landscape corresponds to a full combinatorial multiallelic fitness landscape of 640 engineered mutants of 13 amino-acid changing mutations at six sites in the protein Hsp90 in Saccharomyces cerevisiae [13]. The corresponding fitness landscape was obtained under a condition of elevated salinity, such that the wild-type is no longer the global maximum of the resulting fitness landscape. Figure 1 presents the subset of the genotype space that corresponds to those 640 possible combinations. The nodes represent the sequences whereas the directed links denote the one-mutational transitions that are allowed by the dynamics.

The second empirical fitness landscape encompasses all variants at four amino-acid sites ($20^4 =$ 160 000) in a given region of the protein Gb1 of the Streptococcal bacteria [15]. Additionally, we make a thorough statistical analysis of the properties of adaptive walks for the NK landscape model and three different degrees of epistasis, namely $K = 1$, $K = 2$ and $K = 3$. In this case, several independent landscapes are generated and their properties checked out. Most of the quantities considered here are well established in the literature as legitimate measurements of characterization of evolutionary trajectories in natural populations. Notwithstanding, little is known about how those quantities correlate with each other, and how they respond to variations of the fitness landscape of similar global patterns. With this aim, here we quantify the statistical correlations among all variables of characterization of evolutionary trajectories.

# 2. Methods

## 2.1. The NK landscape model

In the NK model, each individual is represented by a sequence of length $N$, $\mathbf{S} = (s_1, s_2, \ldots, s_N)$ with $s_\alpha = 0$, 1, i.e. a binary base is assumed. The other parameter of the model, $K$, determines the degree of correlation between the elements of $\mathbf{S}$, which means that the contribution $\omega_j$ of an element $j$ to the overall fitness $f$ is a function $\omega_j = g(s_j, \prod(s_j))$ that depends on the state of locus $j$ and on the state of a set of $K$ neighbours, $\prod(s_j)$, that are randomly chosen among the remaining $N - 1$ elements. In this way, the genetic architecture here considered is that of random neighbours, which implies that the fitness landscape here considered does not have a modular structure [38–40]. The values of the components $\omega_j$ are drawn from a uniform distribution (0, 1], and the fitness of the sequence is given by

$$f = \frac{1}{N} \sum_{j=1}^{N} \omega_j. \tag{2.1}$$

The sequence $\mathbf{S}$ may be seen as the genetic code of an individual, and the elements $s_\alpha$ can be considered as its genes. In this perspective, the model is built to allow epistatic interactions among the genes, which means that the contribution of each gene to the overall fitness depends not only on its state but also on the state of $K$ other genes. Because of that, a lookup table for each gene consists of $2^{K+1}$ entries. When $K = 0$ the fitness landscape is simply additive, exhibiting a single peak, and a null level of roughness. In the other extreme, when $K = N - 1$, the landscape becomes completely uncorrelated, and any point mutation affects the contribution of all genes to the sequence's fitness. The case $K = N - 1$ corresponds to the random-energy model [41]. Thus, the parameter $K$ settles the degree of roughness of the fitness landscape. As $K$ increases, so does the number of local maxima of the fitness landscape. Here, the epistatic interactions are randomly assigned, i.e. for each locus, the $K$ loci that epistatically influence its contribution are randomly chosen among the $N$ loci that comprise the genome.

## 2.2. Characterization of the adaptive trajectories

For the empirical fitness landscapes, the adaptive walks are initiated from the wild-type sequence (QFGWSANME for the Hsp90 protein, VDGV for the Gb1 protein), while for the NK landscapes, the adaptive walks are always initiated from the lowest fitness sequence. The trajectories terminate when a local optimum of the landscape is reached. In the first version of the walk, the population (walker) moves to a randomly chosen neighbouring node among those with higher fitness, and so selective differences are not considered. As aforementioned, this version is referred to as random adaptive walk, whereas in the second version, the probabilistic adaptive walk, the move is also random but selective differences are now considered (see equation (1.1)).

We perform a fine-grained analysis of the adaptive walks. For each run, we keep track of the entire array of nodes visited ($S_0, S_1, \ldots, S_f$), which begins at a node $S_0$ (wild-type sequence for empirical landscapes, and least adapted sequence for the NK landscapes) and ends up at one of the local optima of the fitness landscape, $\mathbf{x}_k^{max}$. The dataset for the Hsp90 fitness landscape was built assuming substitutions at the aminoacid level and considering a subset of 13 amino-acid changing mutations [13]. The Hsp90 fitness landscape presents 6 local maxima, all of which are accessible from the wild-type sequence. Note that the wild-type sequence is not a local maxima of the fitness landscape in the high salinity environment. On the other hand, the data for the Gb1 fitness landscape presents a

complete graph comprising all variants at four amino-acid sites, leading to $20^4 = 160\,000$ sequences [15]. Considering that the adaptive moves occur at the aminoacid level, the fitness landscape encompasses 30 local maxima. Of those, fifteen have fitness lower than the wild-type and are clearly ruled out, and the other three local maxima are not accessible from the wild-type at all, meaning that there exists no monotonic fitness paths connecting them to the wild-type. As a consequence, we will focus our attention on the remaining 12 local maxima that are accessible from the wild-type sequence and have at least one monotonic path between them.

The whole set of adaptive walks is split into different subsets according to the terminal point. For instance, for the Hsp90 fitness landscape, the whole set of mutational pathways is divided into six subsets, whereas for the Gb1 fitness landscape twelve distinct subsets are generated. The current approach allows a fine-grained analysis of the properties of the walks but also a deeper insight about the basin of attraction of the local maxima. The same procedure is adopted for the NK landscapes.

In this manner, the evolutionary pathways inside each subset correspond to an ensemble of trajectories in which the starting and ending points are always the same, hence endorsing us to make use of statistical measurements that are already established in the literature to outline those trajectories [12,42–44]. One of those quantities, known as predictability, is defined as

$$P_2^k = \sum_{q_\alpha^k} O^2(q_\alpha^k), \tag{2.2}$$

where $O(q_\alpha^k)$ is the probability of occurrence of the trajectory $q_\alpha^k$ belonging to the ensemble of trajectories that initiates at the wild-type sequence (or least adapted sequence) and terminates at the local maxima $k$. The sum is taken over all distinct trajectories from the trajectory ensemble of size $\mathcal{A}^k$ that ends up at the local optimum $k$. The predictability is just a simple measure of repeatability of the trajectories and can vary from $P_2^k = 1/\mathcal{A}^k$, when all trajectories in the ensemble are disparate, to $P_2^k = 1$, whenever a single trajectory between the wild-type sequence and local optimum $k$ exists [12]. Accordingly, its reciprocal $1/P_2$ gives an estimate of the number of effective trajectories exploited by dynamics.

Note that similar but not exactly identical trajectories contribute with the same weight to the predictability, $P_2^k$, as two completely distinct trajectories. In this sense, completementary information can be gathered by assessing the mean path divergence originally proposed in the study of evolutionary paths by Lobkovsky et al. [43]. The divergence between any pair of trajectories $q_\alpha^k$ and $q_\beta^k$, denoted by $d(q_\alpha^k, q_\beta^k)$, is computed by measuring the Hamming distance of each sequence in $q_\alpha^k$ to all sequences in $q_\beta^k$, hence storing the shortest distance only. Perceive that the length of the two trajectories do not need to be the same. As follows, $d(q_\alpha^k, q_\beta^k)$ is then defined as the average of these shortest Hamming distances over all strings in $q_\alpha^k$. Finally, the mean path divergence is calculated as

$$\bar{d}^k = \sum_{q_\alpha^k} O(q_\alpha^k) \sum_{q_\beta^k} O(q_\beta^k) d(q_\alpha^k, q_\beta^k), \tag{2.3}$$

which yields the expected divergence of two trajectories drawn at random from the ensemble of trajectories [43].

It is important to emphasize that both quantities, $P_2^k$ and $\bar{d}^k$, are only meaningful provided that the starting and ending points of the adaptive walks are always the same within an ensemble of trajectories, a condition that is met in our approach.

Other statistical quantities also assessed through the adaptive walks are the accessibility of the local optima and mean walk lengths. In addition, for each fitness landscape a graph analysis is carried out. In that analysis, a graph accessibility of a given local optimum is defined, corresponding to the ratio between the number of pathways connecting the starting node $S_0$ to the focal local maximum and the total number of pathways departing from $S_0$. In the way it is defined, the graph accessibility gives the same weight for every path, meaning that all paths are equally likely with no regards to their lengths. For the Hsp90 and the NK landscapes, the whole set of all possible monotonic pathways connecting the starting point and the local maxima of the landscapes are generated and their properties examined. For the Gb1 landscape, a complete graph analysis could not be undertaken because of the extremely high computational and time costs involved. This problem arises as a consequence of the high connectivity of the graph, each node has coordination $z = 20$, leading to very long mutational pathways. For the Gb1, the graph accessibility is indirectly inferred from an ensemble of trajectories of the random adaptation walks, since it is expected that in the limit of an infinitely large number of adaptive walks, all accessible paths can be reached. In the following, we present a summary of the statistical properties here assessed and their abbreviations.

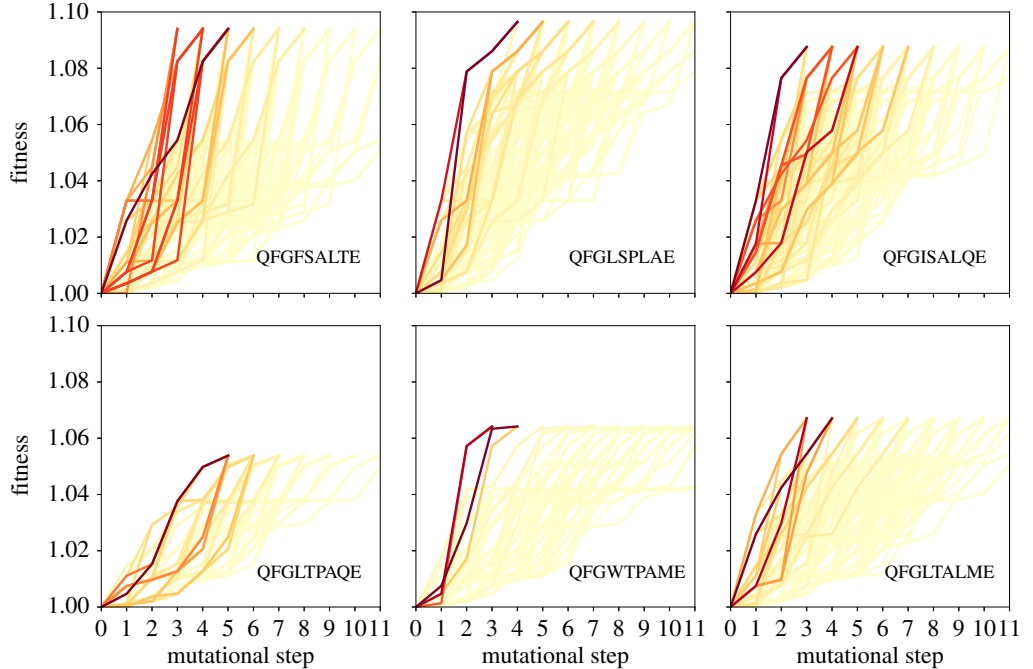

**Figure 2.** The set of all monotonic mutational pathways from the wild-type sequence (QFGWSANME) to the six local maxima of the Hsp90 fitness landscape. The corresponding target sequences are QFGFSALTE, QFGLSPLAE, QFGISALQE in the top row and QFGLTPAQE, QFGWTPAME, QFGLTALME in the bottom row. QFGLSPLAE is the global optimum of the Hsp90 fitness landscape. A colour gradient has been used to denote the frequency at which the paths are assessed through the dynamics. Data obtained from **random** adaptive walks. The darkest lines represent the most assessed mutational pathways.

## 2.3. Summary of the quantities assessed

In this section, we present the statistical measurements obtained from the three different approaches used in the paper, which are:

—random adaptative walks:
   —**mean path rand:** mean walk length for random adaptive walks
   —**access rand:** accessibility under random adaptive walks
   —**predi rand:** predictability under random adaptive walks
   —**diver rand:** mean path divergence under random adaptive walks
—probabilistic adaptive walks:
   —**mean path prob:** mean walk length for probabilistic adaptive walk
   —**access prob:** accessibility under probabilistic adaptive walk
   —**predi prob:** predictability under random adaptive walks
   —**diver prob:** predictability under probabilistic adaptive walks
—graph analysis:
   —**mean path graph:** average path length of all paths connecting $S_0$ to the local maxima
   —**access graph:** accessibility from the perspective of the graph analysis
   —**min path graph:** minimum distance from the starting point and the local maxima
   —**max path graph:** maximum distance from the starting point and the local maxima

# 3. Results and discussions

In figures 2 and 3, a graph-based metadata analysis provides all monotonic pathways from the wild-type sequence to the six local maxima of the Hsp90 fitness landscape under random and probabilistic dynamics, respectively. A colour gradient is employed to highlight those paths that are most frequently assessed through the adaptive walks. The results for both dynamics, random and probabilistic, are shown. We observe that the most operated trajectories for random and the probabilistic dynamics exhibit a great overlapping. In general, we observe that the shortest paths are those more regularly used, regardless of the dynamics. While some local optima display a huge number of alternative monotonic

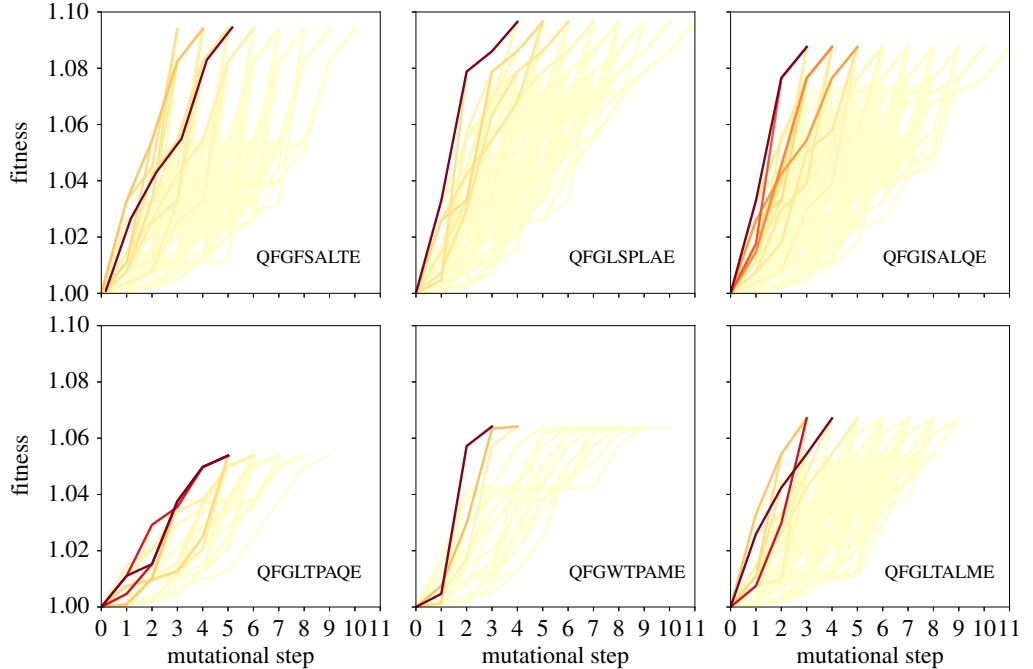

**Figure 3.** The set of all monotonic mutational pathways from the wild-type sequence (QFGWSANME) to the six local maxima of the Hsp90 fitness landscape. The corresponding target sequences are QFGFSALTE, QFGLSPLAE, QFGISALQE in the top row and QFGLTPAQE, QFGWTPAME, QFGLTALME in the bottom row. QFGLSPLAE is the global optimum of the Hsp90 fitness landscape. A colour gradient has been used to denote the frequency at which the paths are assessed through the dynamics. Data obtained from **probabilistic** adaptive walks. The darkest lines represent the most assessed mutational pathways.

pathways, the number of possibilities is considerably reduced for other ones, as noted for the local maximum QFGLTPAQE. In fact, as the walker endures longer distances, the number of possibilities increases [45]. Although the number of ramifications will not grow indefinitely, owing to epistasis effects, it is maximized at intermediate distances. This explains why short-length pathways are usually promoted in detriment of the long length pathways, even for random adaptive walks. An additional reason for the frequent usage of short-length pathways upon probabilistic adaptive walks is that short-length walks are exactly those which are steeper, requiring that the selective effects along the pathway are reasonably large, and so more often singled out. All of these will certainly impact the predictability of the evolutionary dynamics. Similar patterns are found for the Gb1 landscape, and the theoretical NK landscape model (see electronic supplementary material). Some of the mutational pathways are indirect monotonic pathways, whose lengths are larger than the hamming distance from the wild-type to the local optimum, and pass through nodes that are even farther from the wild-type sequence. Electronic supplementary material, figure S1 exhibits the most used paths for the Gb1 landscape under the probabilistic dynamics, and electronic supplementary material, figures S2–S3 display all accessible paths for a single instance of the NK landscape.

The accessibility of the local maxima of the Hsp90 (upper panel) and Gb1 (lower panel) fitness landscapes is shown in figure 4. In the plot, the sequences are arranged in ascending order of fitness. While the accessibility of the local maxima through the dynamics seems to be reasonably correlated with the graph accessibility, the two quantities are not exactly proportional. For both landscapes, one sees some occurrences of local maxima that present high graph accessibility but are relatively less accessed through the dynamics. We also note that while for the Hsp90 fitness landscape both accessibility and graph accessibility tends to increase with the fitness of the local optima, such a pattern is less clear for the Gb1 fitness landscape. Indeed, the relation between accessibility and fitness is still not yet clarified. For uncorrelated fitness landscapes, it is expected that accessibility increases with the fitness difference between the starting and target sequences, and decreases with sequence size [45]. In order to more deeply address this issue and try to draw more general inferences, the panels of figure 5 display the accessibility and graph accessibility for the NK fitness landscape model. In the panels, the sequence size is $N = 8$ and different degrees of epistasis are considered: $K = 1$ (upper panel), $K = 2$ (middle panel), and $K = 3$ (lower panel). By changing $K$, the level of fitness correlation

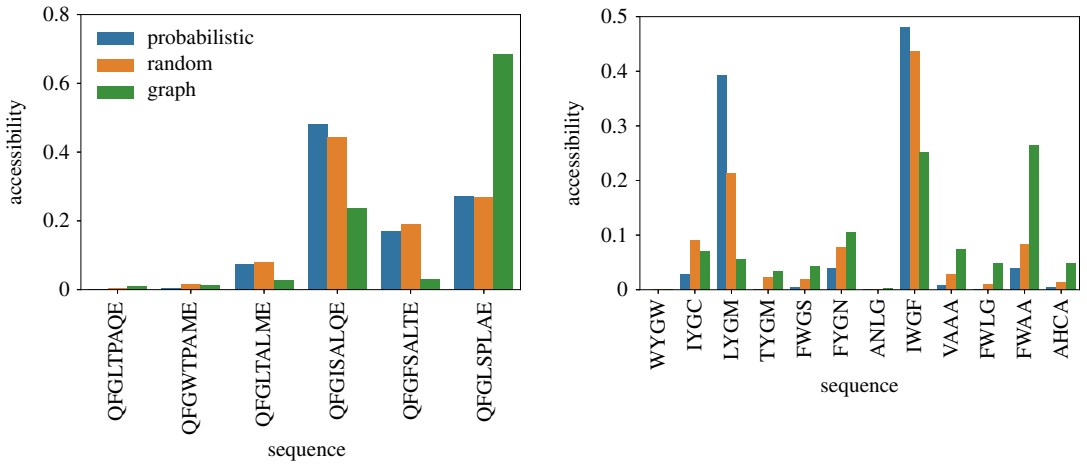

**Figure 4.** Accessibility of the local optima of the empirical fitness landscapes Hsp90 (upper panel) and Gb1 (lower panel). The orange and blue bars denote the results for the random and probabilistic adaptive walks, respectively. Whereas, the green bars are the graph accessibility, defined as fraction of monotonic mutational pathways that depart from the starting node $S_0$ and terminate at the respective local optimum.

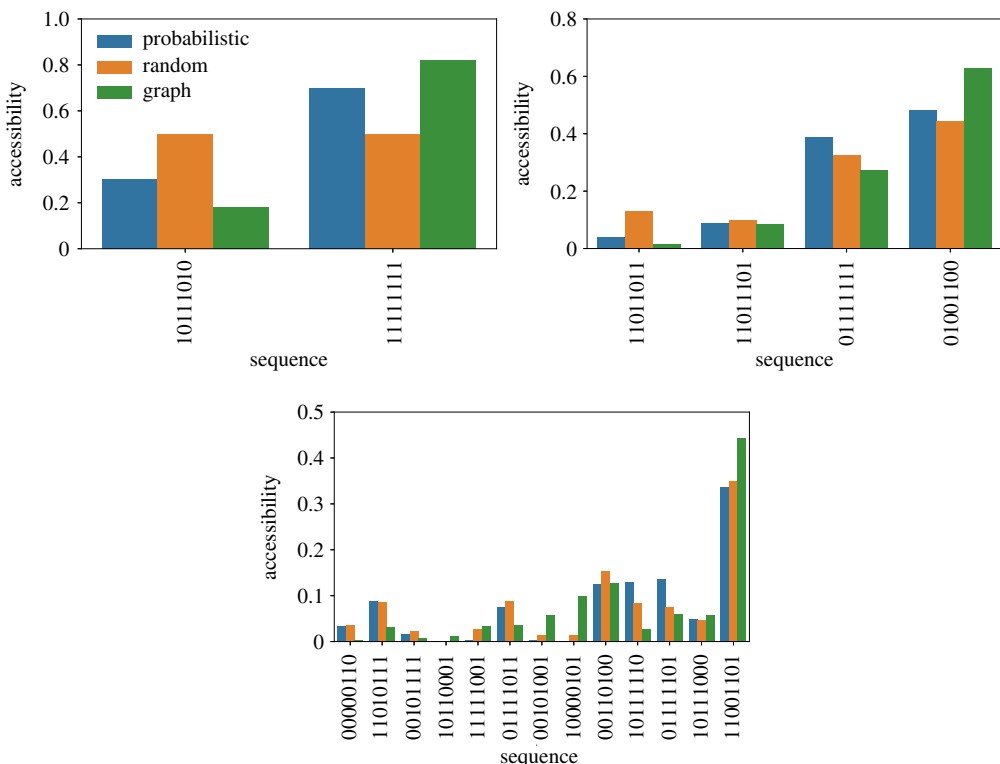

**Figure 5.** Accessibility of the local optima of the NK-model of fitness landscapes for $N = 8$, and epistasis parameter $K = 1$, 2 and 3, from upper to lower panels, respectively. The bar colours follow the convention presented in figure 4.

among one-step neighbours $\rho = K + 1/N$ changes and reaches $\rho \approx 0.5$ for $K = 3$. It is important to highlight that those results presented in figure 5 for the NK model refer to a single instance of the fitness landscape. By examining the panels, we observe that there exists a clear correlation between accessibility, as defined by dynamics, and graph accessibility. Nonetheless, while this association between the two variables seems to be quite tight when $K = 1$ and $K = 2$, it becomes less clear for $K = 3$.

Next, we check out two quantities that have been used to characterize the distribution of accessible pathways [12,43,44]. Figures 6 and 7 show the predictability, as given by equation (2.2), and figures 8 and 9 show the mean path divergence, defined by equation (2.3), for the ensemble of trajectories for each local optimum of the Hsp90 and Gb1 landscapes, as well as the same NK fitness landscapes analysed in the previous figure. By comparing these figures and looking more closely at the extreme

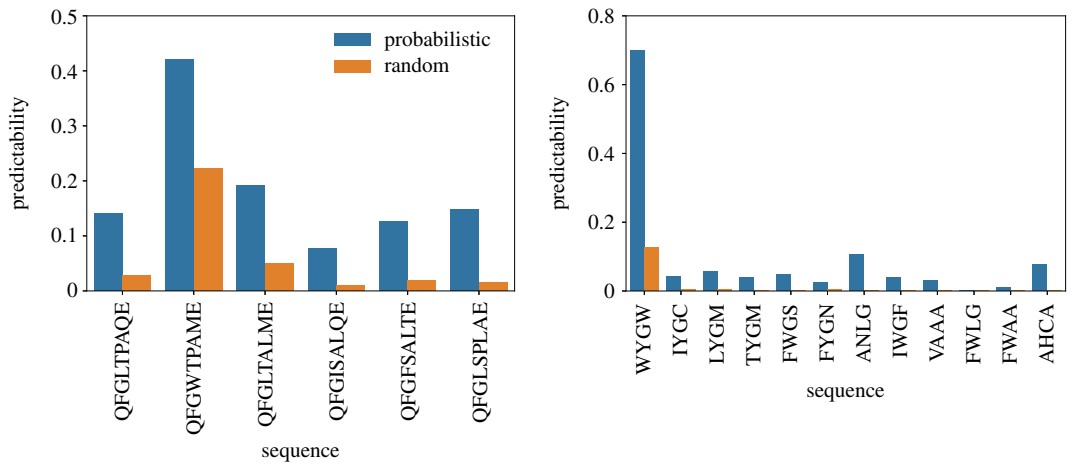

**Figure 6.** Predictability of the evolutionary trajectories followed by the adaptive walks initiated at the wild-type to each accessible local optimum of the empirical fitness landscapes Hsp90 (upper panel) and Gb1 (lower panel). The orange and blue bars denote the results for the random and probabilistic adaptive walks, respectively.

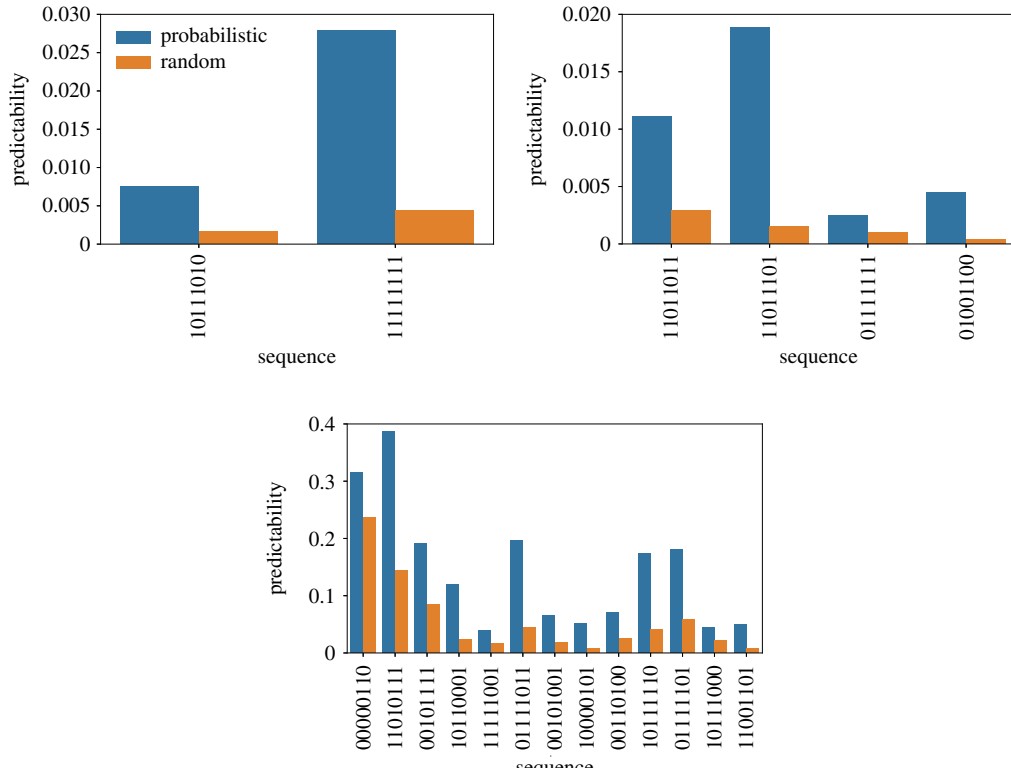

**Figure 7.** Predictability of the evolutionary trajectories followed by the adaptive walks initiated at the least adapted sequence to each accessible local optimum of the NK-model of fitness landscape for $N = 8$. From top to bottom panels, the epistasis parameters are $K = 1$, 2 and 3, respectively. The bar colours follow the scheme of figure 6.

cases, it is possible to guess the occurrence of a negative correlation between predictability and mean path divergence. The predictability $P_2$ seems to be a much more responsive quantity than the mean path divergence, which exhibits a relatively smaller variation. Although, at least for the extreme cases, we observe for both Hsp90 and Gb1 landscapes that the highest predictability values go along with the smallest mean path divergence values, while the reverse situation also applies. In between, we still observe an overall tendency of decreased mean path divergence as the predictability grows.

While it is still apparent that there is a negative correlation between accessibility and predictability for the Hsp90 fitness landscape, when one contrasts figures 4 and 6, one can not easily infer the interdependence between the two quantities when resolving the data for the Gb1 fitness landscape.

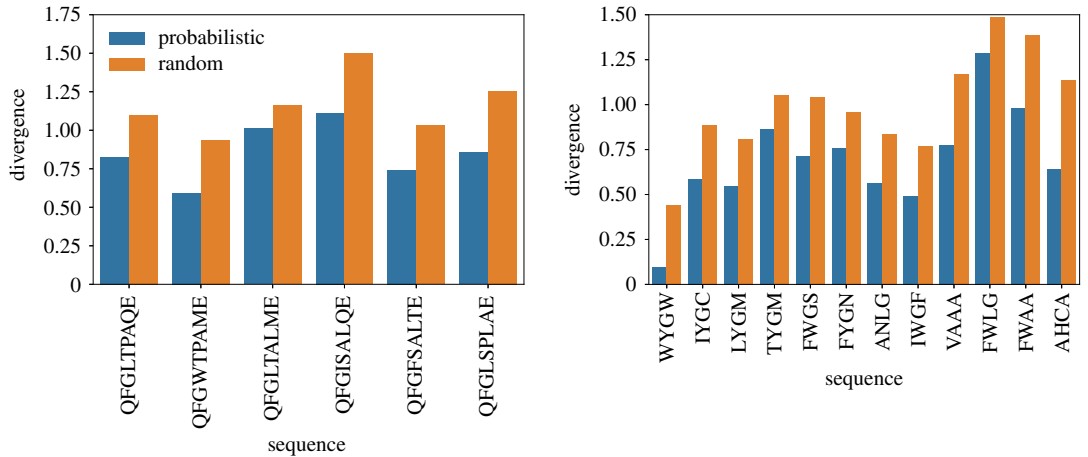

**Figure 8.** Mean path divergence of the ensemble of mutational pathways for each accessible local optimum of the empirical fitness landscapes Hsp90 (upper panel) and Gb1 (lower panel). The orange bars represent the results for the random adaptive walks and blue bars denote the results for the probabilistic ones.

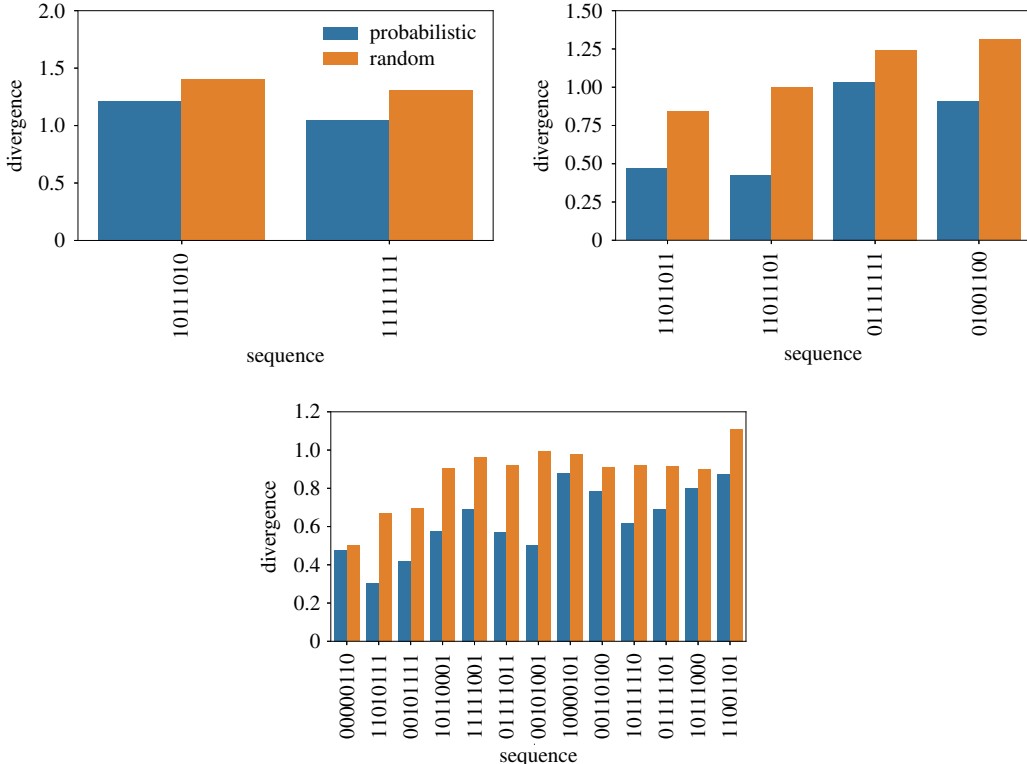

**Figure 9.** Mean path divergence of the ensemble of mutational pathways for each accessible local optimum for single instances of NK fitness landscapes. The sequence size is $N = 8$, whereas the epistasis parameters are $K = 1$, 2 and 3 (from top to bottom panels). The colour bars follow the scheme presented in figure 8.

For the NK fitness landscapes, the situation is even more troublesome, as the association between accessibility and predictability seems to depend on the level of epistasis. The results for $K = 1$ suggest a positive association between the two variables, while the contrary seems to occur for $K = 2$ and $K = 3$. Remember that these data concern a single instance of the fitness landscape. A great convenience of dealing with theoretical landscapes is that any hypothesis can be tested over a set of independently generated fitness landscapes in a tunable way.

Theoretical fitness landscapes, such as the NK model, can provide productive insights into this problem as an ensemble of fitness landscapes can be generated while their global properties are preserved, such as the level of epistasis and fitness correlation among one-step neighbours.

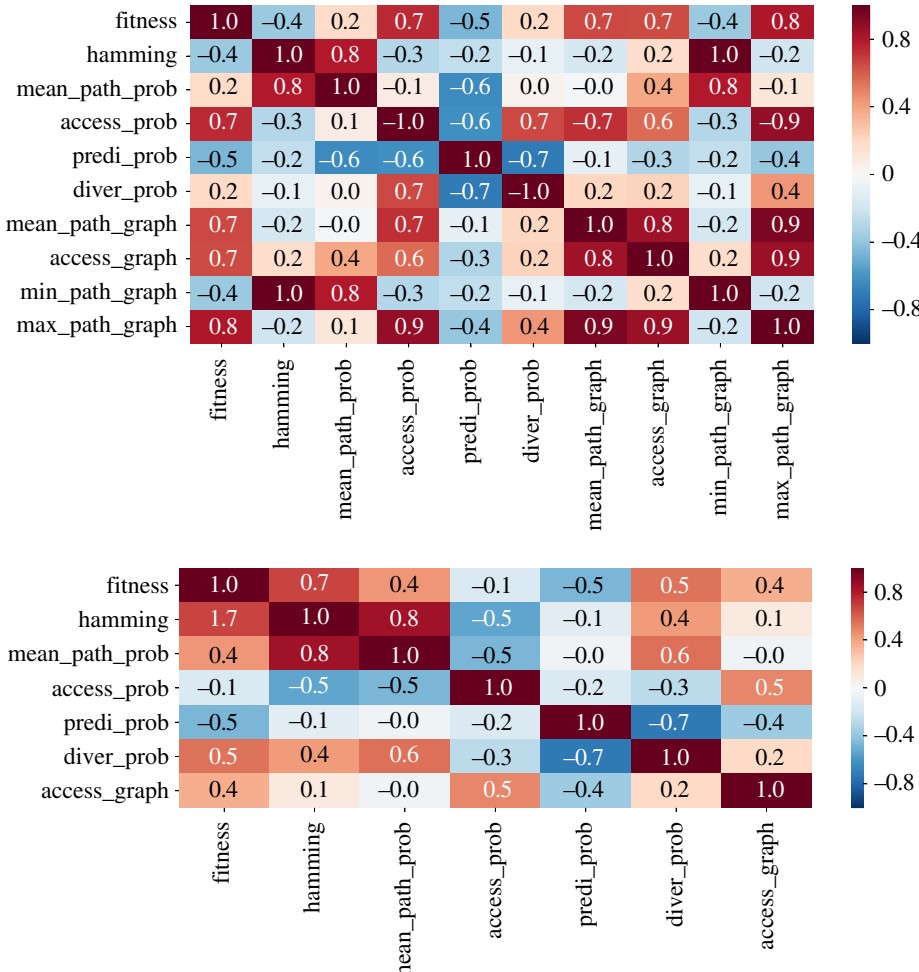

**Figure 10.** Correlation tables specifying the level of correlation between probabilistic adaptive walks and graph measures of the empirical landscapes Hsp90 in the upper panel and the Gb1 in the lower panel. The abbreviations of the variables in the label are declared in §2.3.

Additionally, an ensemble of fitness landscapes can help us in gathering information on the variability and spreading of the resulting correlation coefficients for pairwise comparisons between the quantities of interest. This will be done in the following.

In order to better synthesize the aforementioned outcomes, figure 10 presents a correlation table, which specifies the level of statistical correlation between a pair of variables. The results refer to the probabilistic adaptive walk and concern the empirical fitness landscapes, Hsp90 and Gb1. The table derived for the Gb1 fitness landscape does not include the quantities assessed through the graph analysis for the reasons raised previously, with the exception of the graph accessibility, which in this case is indirectly estimated from the random adaptive dynamics. On the other hand, figure 11 presents the correlation tables of NK landscapes and for different levels of epistasis. In this case, ten independent replicates of the fitness landscape for each value of the epistasis parameter $K$ were assembled for evaluation of the average values. Electronic supplementary material, figures S6–S8 plots the correlation tables of the individual fitness landscapes used to obtain the final results displayed in figure 11.

Electronic supplementary material, figures S6–S8 disclose an impressive variation of the statistical correlation values across the distinct samples of fitness landscapes, while keeping the degree of epistasis $K$. Some of the pairwise correlation coefficients change not only their strength but even their sign as other landscapes are analysed. While keeping in mind that very different scenarios may come about when comparing distinct samples of fitness landscapes, the results shown in figure 11 provide a robust and important guide for understanding how those are linked to each other.

Next, we proceed with the analysis of the information gathered from all those correlation tables. A first point that becomes clear from this essay is that while graph accessibility is positively correlated to the accessibility of the fitness peaks via the dynamics, the relation seems to be weakened with

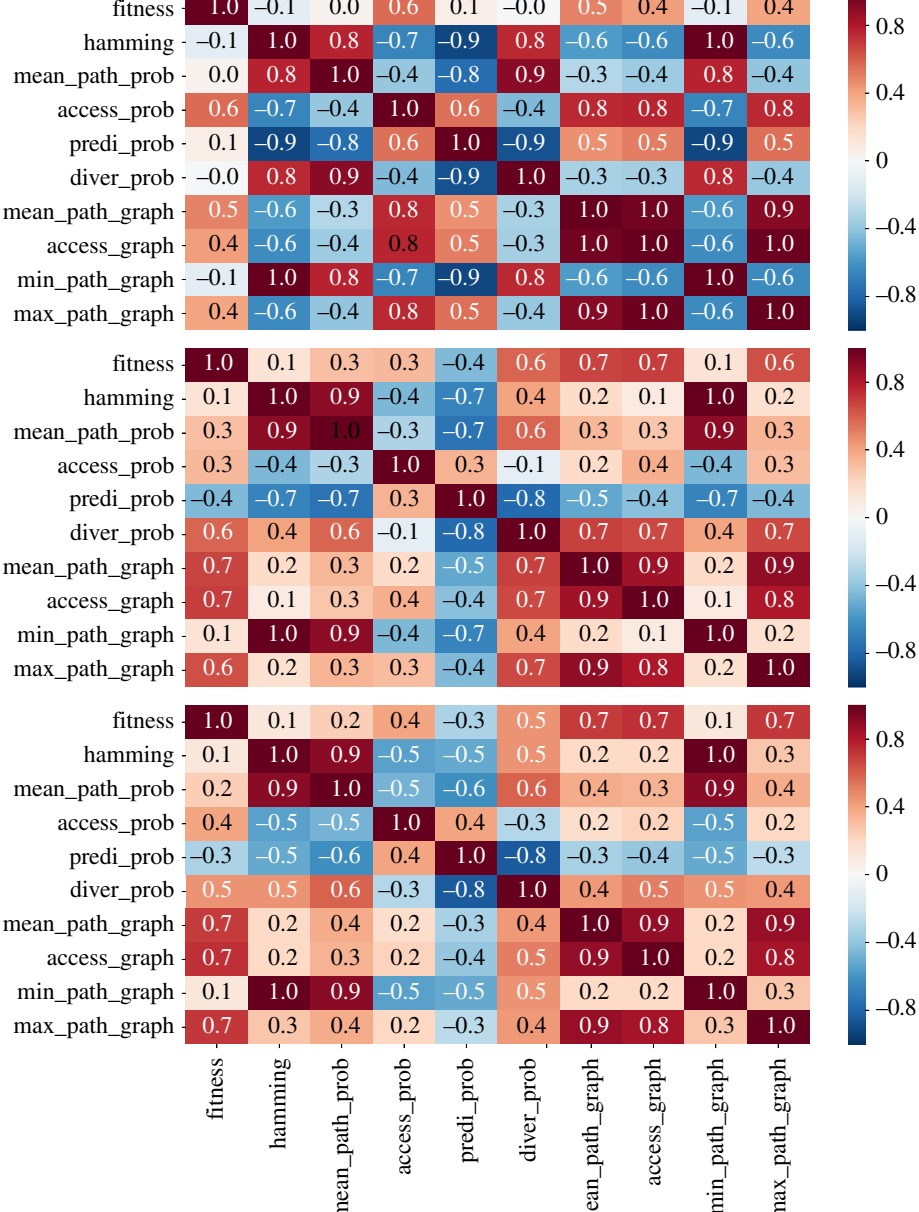

**Figure 11.** Correlation between probabilistic adaptive walks and graph measures of the NK-model with $N = 8$ and epistasis degrees of $K = 1$ (upper panel), $K = 2$ (middle panel) and $K = 3$ (lower panel). The abbreviations of the variables in the label are declared in §2.3.

increased ruggedness, $K$, of the fitness landscape. The prediction for the empirical landscapes seems to be compatible with those of the NK landscapes with a reasonable amount of epistasis. This outcome suggests that as the landscape becomes more rugged, graph accessibility becomes less informative about the basin of attraction of the set of local optima, possibly owing to an increased number of nodes belonging to the basin of attraction of distinct local optima.

Other relations are of special interest, especially those widely employed in the characterization of adaptive walks, such as the mean walk length, hamming distance and fitness values. In general, we observe a negative correlation between the accessibility of the fitness peaks and the mean walk length, which is a direct consequence of the negative correlation between accessibility and hamming distance, i.e. the local maxima that are more easily reached are exactly those whose hamming distance is small. In spite of still playing a role, the height of the fitness peaks is less influential in determining the walk lengths than their hamming distance to the wild-type.

Concerning the predictability and the mean path divergence, we observe that in general the mean walk length is negatively correlated to the predictability but positively associated with the mean path divergence. In a coarse-grained description, one can say that the negative correlation between mean

walk length and predictability is a consequence of the fact that longer walks encompass more possible combinations of paths and additionally are associated with less steep trajectories. On the other hand, the positive correlation between mean walk length and mean path divergence follows from the positive association between walk lengths and hamming distance. Mainly, long walks occur for those local optima which are further away, which in turn opens new room for the paths to cover a broader and divergent domain across the genotype space. These combined effects result in a strong negative correlation between predictability and mean path divergence, which is verified in all scenarios.

Except for the Gb1 fitness landscape, there exists a positive correlation between the fitness of the local maxima and accessibility. This outcome evinces the finding that higher peaks of fitness landscapes are in general more accessible, as demonstrated for uncorrelated fitness landscapes [45], and has greater applicability. On the other hand, negative values of correlation are found between fitness values of the peaks and their corresponding measures of predictability, over nearly all scenarios. Indeed, the larger the fitness difference between the start and end points of the adaptive walks, more alternative routes that embrace an increased number of mutations of small effect arise, thus contributing to longer walks, and increased unpredictability.

# 4. Concluding remarks

The concept of the fitness landscape is central in evolutionary biology, and recent efforts have been concentrated on surveying how the topography of fitness landscapes steers evolution [16,17]. The topography of fitness landscapes is mainly determined by epistatic interactions among genetic loci [12,46]. Single-peak and smooth landscapes are characterized by a lack of epistasis. On the other hand, epistatic interactions introduce curvature, and the occurrence of sign and reciprocal sign epistasis can result in multi-peaked landscapes [47,48]. In smooth landscapes, the outcome of the evolutionary process is quite predictable, whereas the number of mutational pathways is vast. By contrast, in rugged landscapes one loses the power of predicting the evolutionary outcome, nevertheless, the mutational pathways through sequence space are constrained by sign epistasis, and so the direction of evolution can be more easily anticipated [49,50].

A more detailed description of evolutionary processes requires us to gather information at the genetic level, i.e. by storing the set of genotypes visited along the adaptive process [43,44]. Within this scenario, new measurements were proposed in order to characterize the distribution and frequency of mutational pathways. Therefore, it is crucial to understand how those measurements are associated and how susceptible they are to variations of the fitness landscape. This is the main objective of the present study.

Here, we performed an exhaustive analysis of the statistical properties of the adaptive walks in two empirical fitness landscapes [13,15]. Their results are compared with those obtained from theoretical NK landscapes. The importance of dealing with theoretical landscapes is that an ensemble of fitness landscapes can be produced while global properties are preserved. The analysis of theoretical landscapes allowed us to conclude that the variation obtained in the characterization of the empirical landscapes are compatible with the observed variation in NK landscapes. Some general conclusions can be drawn: positive correlations are established between graph accessibility and accessibility through the dynamics; accessibility and fitness; mean walk length and mean path divergence; whereas negative correlations are established between predictability and mean path divergence; accessibility and Hamming distance; mean walk length and predictability. On the other hand, the pairwise correlation between accessibility and predictability is not preserved across fitness landscapes. While for the two empirical fitness landscapes the two quantities are inversely correlated, the opposite behaviour is found for NK landscapes.

Data accessibility. The code and data can be downloaded from the Dryad Digital Repository: doi:10.5061/dryad.41ns1rn9r https://datadryad.org/stash/share/aQWKqzRIXXVjsOLBBEZvbHhuVHvxgcDYLq9VO24ew1Q.
Authors' contributions. P.R.A.C. designed the study, carried out computer simulations of the adaptive walks, participated in data analysis and drafted the manuscript; S.M.R. compiled the data from the empirical fitness landscapes, performed a graph analysis of the evolutionary trajectories, participated in data analysis and drafted the manuscript.
Competing interests. The authors declare no competing interests.
Funding. P.R.A.C. is supported in part by Conselho Nacional de Desenvolvimento Científico e Tecnológico (CNPq)by grant nos 302569/2018-9 and 406594/2018-0, and Edital QUALIS A (UFPE). S.M.R. is supported by the Programa Nacional de Pós-Doutorado/Capes (PNPD/CAPES).
Acknowledgements. The authors acknowledge J. F. Fontanari for helpful insights and fruitful discussions. The authors also acknowledge Claudia Bank for providing the data of the Hsp90 fitness landscape.

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
