## [Reviewer comments · Royal Society Open Science]

Review History

RSOS-192118.R0 (Original submission)

Review form: Reviewer 1

Is the manuscript scientifically sound in its present form?

Yes

Are the interpretations and conclusions justified by the results?

Yes

Is the language acceptable?

Yes

Do you have any ethical concerns with this paper?

No

Have you any concerns about statistical analyses in this paper?

No

Recommendation?

Accept with minor revision (please list in comments)

Comments to the Author(s)

The authors presented a much improved version of their manuscript. The abstract reads much better e describes the main results in a much clearer way. Figures and correlation tables have also been improved and are easier to visualise and understand.

I believe the authors answered all the points raised by me and the other reviewer in a satisfactory way and I now suggest its approval for publication.

I have only a few minor points only:

Abstract:

Some general scenario can be drawn -> Some general conclusions ??

A negative correlation predictability -> A negative correlation between predictability

and so with the decrease -> and so the decrease

The description of the NK model is much better but I confess I had to look it up in wikipedia to understand it. My confusion is that you say that the ω_j are taken from a uniform distribution and the fitness is the average of these ω_j . Well, this looks like $1/2!$ The point is that what is taken from a uniform distribution are not the ω_j but the values of the functions. So I suggest (it is only a suggestion) that they use

$$\omega_j = g(s_j, \prod(s_j))$$

and say that the values of the function g , calculated at its several arguments, is taken from a random uniform distribution.

Review form: Reviewer 2

Is the manuscript scientifically sound in its present form?

Yes

Are the interpretations and conclusions justified by the results?

Yes

Is the language acceptable?

Yes

Do you have any ethical concerns with this paper?

No

Have you any concerns about statistical analyses in this paper?

No

Recommendation?

Accept with minor revision (please list in comments)

Comments to the Author(s)

The paper contains a number of small grammatical mistakes; these are most notable in the abstract/introduction. Examples:

Abstract:

- > "adaptive walks are an idealized dynamics" should be "are an idealized dynamic" or "are idealized dynamics that mimic"
- > "some general scenario" should be "some general scenarios"
- > "regardless the dynamics" should be "regardless of the dynamics"
- > "a negative correlation predictability" should be "a negative correlation between predictability"
- > "and so with the decrease of the number of effective mutational pathways" should be "and so a decrease in the number of"

I recommend the authors go through the introduction and conclusion section with a careful eye for English grammar, but find the overall language and conclusions of the paper acceptable, and appreciate the authors' responses to my and the other reviewer's comments

Decision letter (RSOS-192118.R0)

19-Dec-2019

Dear Dr Campos,

On behalf of the Editors, I am pleased to inform you that your Manuscript RSOS-192118 entitled "Analysis of statistical correlations between properties of adaptive walks in fitness landscapes" has been accepted for publication in Royal Society Open Science subject to minor revision in accordance with the referee suggestions. Please find the referees' comments at the end of this email.

The reviewers and handling editors have recommended publication, but also suggest some minor revisions to your manuscript. Therefore, I invite you to respond to the comments and revise your manuscript.

- Ethics statement

- Data accessibility

<http://datadryad.org/submit?journalID=RSOS&manu=RSOS-192118>

- **Competing interests**

- **Authors' contributions**

- **Acknowledgements**

- **Funding statement**

Because the schedule for publication is very tight, it is a condition of publication that you submit the revised version of your manuscript before 28-Dec-2019. Please note that the revision deadline will expire at 00.00am on this date. If you do not think you will be able to meet this date please let me know immediately.

1) Identifying all the changes that have been made (for instance, in coloured highlight, in bold text, or tracked changes);

If your manuscript is newly submitted and subsequently accepted for publication, you will be asked to pay the article processing charge, unless you request a waiver and this is approved by Royal Society Publishing. You can find out more about the charges at <https://royalsocietypublishing.org/rsos/charges>. Should you have any queries, please contact openscience@royalsociety.org.

on behalf of the Associate Editor and Professor Kevin Padian (Subject Editor)
openscience@royalsociety.org

Associate Editor Comments to Author:

Thank you for submitting this revision - it seems a few remaining scientific tweaks are necessary, but possibly the most pertinent is a final language check to improve clarity. You might benefit from the advice of a service such as those available at <https://royalsociety.org/journals/authors/language-polishing/> - alternatively, you might consider reaching out to a colleague who is a native speaker of English for advice (we recognise the language is a sometimes peculiar beast, and do sympathise). We'll look forward to receiving your final revision.

Reviewer comments to Author:

Reviewer: 1

Comments to the Author(s)

The authors presented a much improved version of their manuscript. The abstract reads much better e describes the main results in a much clearer way. Figures and correlation tables have also been improved and are easier to visualise and understand.

I believe the authors answered all the points raised by me and the other reviewer in a satisfactory way and I now suggest its approval for publication.

I have only a few minor points only:

Abstract:

Some general scenario can be drawn -> Some general conclusions ??

A negative correlation predictability -> A negative correlation between predictability

and so with the decrease -> and so the decrease

The description of the NK model is much better but I confess I had to look it up in wikipedia to understand it. My confusion is that you say that the ω_j are taken from a uniform distribution and the fitness is the average of these ω_j . Well, this looks like $1/2!$ The point is that what is taken from a uniform distribution are not the ω_j but the values of the functions. So I suggest (it is only a suggestion) that they use

$$\omega_j = g(s_j, \prod(s_j))$$

and say that the values of the function g , calculated at its several arguments, is taken from a random uniform distribution.

Reviewer: 2

Comments to the Author(s)

The paper contains a number of small grammatical mistakes; these are most notable in the abstract/introduction. Examples:

Abstract:

> "adaptive walks are an idealized dynamics" should be "are an idealized dynamic" or "are idealized dynamics that mimic"

- > "some general scenario" should be "some general scenarios"
- > "regardless the dynamics" should be "regardless of the dynamics"
- > "a negative correlation predictability" should be "a negative correlation between predictability"
- > "and so with the decrease of the number of effective mutational pathways" should be "and so a decrease in the number of"

I recommend the authors go through the introduction and conclusion section with a careful eye for English grammar, but find the overall language and conclusions of the paper acceptable, and appreciate the authors' responses to my and the other reviewer's comments

Author's Response to Decision Letter for (RSOS-192118.R0)

See Appendix A.

Decision letter (RSOS-192118.R1)

13-Jan-2020

Dear Dr Campos,

It is a pleasure to accept your manuscript entitled "Analysis of statistical correlations between properties of adaptive walks in fitness landscapes" in its current form for publication in Royal Society Open Science.

Kind regards,
Lianne Parkhouse
Editorial Coordinator
Royal Society Open Science

on behalf of Professor Kevin Padian (Subject Editor)
openscience@royalsociety.org

Appendix A

December 22, 2019

Dear Editor,

First of all, we want to thank you for the positive assessment and acceptance of our manuscript “Analysis of statistical correlations between properties of adaptive walks in fitness landscapes” for publication in Royal Society Open Science subject to minor revision. We would also like to thank the Associate Editor as well as the two reviewers for their very constructive contributions during the peer review process.

We have revised the manuscript and all minor points raised in the reports were considered in this final version. A careful grammar revision of the abstract, introduction and conclusions was carried out.

Yours sincerely,

The authors